# Pulmonary Hypertension and Obesity: Focus on Adiponectin

**DOI:** 10.3390/ijms20040912

**Published:** 2019-02-20

**Authors:** Fabio Perrotta, Ersilia Nigro, Mariano Mollica, Adriano Costigliola, Vito D’Agnano, Aurora Daniele, Andrea Bianco, Germano Guerra

**Affiliations:** 1Department of Medicine and Health Sciences “Vincenzo Tiberio”, University of Molise, Via Francesco De Sanctis, 86100 Campobasso, Italy; v.dagnano@studenti.unimol.it (V.D.); germano.guerra@unimol.it (G.G.); 2Lungs for Living Research Centre, UCL Respiratory, University College London, London WC1E 6JF, UK; e.nigro@ucl.ac.uk; 3Department of Translational Medical Sciences, University of Campania “L. Vanvitelli”/Hosp. Monaldi, 80131 Naples, Italy; mollicamariano@gmail.com (M.M.); adriano.costigliola@studenti.unicampania.it (A.C.); andrea.bianco@unicampania.it (A.B.); 4Dipartimento di Scienze e Tecnologie Ambientali, Biologiche, Farmaceutiche, Università della Campania “Luigi Vanvitelli”, 81100 Caserta, Italy; aurora.daniele@unicampania.it; 5CEINGE-Biotecnologie avanzate, 80145 Naples, Italy

**Keywords:** pulmonary hypertension, adiponectin, obesity, vascular smooth muscle, adipocytokine

## Abstract

Pulmonary hypertension is an umbrella term including many different disorders causing an increase of the mean pulmonary arterial pressure (mPAP) ≥ 25 mmHg. Recent data revealed a strong association between obesity and pulmonary hypertension. Adiponectin is a protein synthetized by the adipose tissue with pleiotropic effects on inflammation and cell proliferation, with a potential protective role on the pulmonary vasculature. Both in vivo and in vitro studies documented that adiponectin is an endogenous modulator of NO production and interferes with AMP-activated protein kinase (AMPK) activation, mammalian target of rapamycin (mTOR), and nuclear factor kappa-light-chain-enhancer of activated B cells (NF-κβ) signaling preventing endothelial dysfunction and proliferation. Furthermore, adiponectin ameliorates insulin resistance by mediating the biological effects of peroxisome proliferator-activated receptor-gamma (PPARγ). Therefore, adiponectin modulation emerged as a theoretical target for the treatment of pulmonary hypertension, currently under investigation. Recently, consistent data showed that hypoglycemic agents targeting PPARγ as well as renin–angiotensin system inhibitors and mineralocorticoid receptor blockers may influence pulmonary hemodynamics in different models of pulmonary hypertension.

## 1. Pulmonary Hypertension: An Overview

Pulmonary hypertension (PH) is a clinical condition characterized by an increase in pulmonary vascular resistance, which may result in heart failure. PH is defined as an increase in the mean pulmonary arterial pressure (mPAP) ≥ 25 mmHg at rest as assessed by right-heart catheterization (RHC) [1,2]. PH hallmarks include medial hypertrophy with an increase in the number and size of pulmonary arterial smooth muscle cells (PASMCs), intimal proliferation, fibrinoid necrosis, and finally, albeit uncommon, plexiform lesions [1]. Factors including individual susceptibility and metabolic status may influence the onset and development of the main changes in pulmonary vessels [3]. Pulmonary Hypertension is categorized into five groups [2]. Group 1, pulmonary arterial hypertension (PAH), comprises the idiopathic and hereditary forms of pulmonary hypertension. The term PAH refers to the presence of pre-capillary PH, defined by a pulmonary artery wedge pressure (PAWP) ≤ 15 mmHg and a pulmonary vascular resistance more than 3 Wood units (WU) in the absence of other causes of pre-capillary PH. This group also includes forms related to the human immunodeficiency virus, liver cirrhosis, connective-tissue disease, and congenital heart disease, as well as drug-induced PAH. Group 2 includes forms of pulmonary hypertension related to left-sided heart disease. Group 3 comprises pulmonary hypertension associated with lung diseases. Group 4 includes chronic thromboembolic disorders associated with chronic PH. In chronic thromboembolic pulmonary hypertension (CTEPH), the presence of organic thrombi causes the increase in pulmonary arterial pressure, with pulmonary vessel remodeling leading to right-ventricular failure. Group 5 includes heterogeneous conditions which are not considered part of the previous groups (sarcoidosis or myeloproliferative disorders). This review examines the current data linking obesity with the different forms of pulmonary hypertension, with a focus on the biological activities mediated by adiponectin.

## 2. Obesity and Pulmonary Hypertension

Over the past three and half decades, the prevalence of obesity has almost doubled worldwide. Among children (less than 18 years of age), 11% of males and 15% of females were obese in 2014. More than 42 million children under the age of five were reported to be overweight in 2013. In adults, 38.3% of women and 34.3% of men in the US are obese [4]. Limited data are available about the prevalence of pulmonary hypertension within the obese population, and most of them arise from retrospective single-center studies. McQuillan et al. reported that 5% of healthy subjects with a body mass index (BMI) > 30 kg/m^2^ had moderate to severe pulmonary hypertension, with a systolic pulmonary arterial pressure (PAPs) greater than 50 mmHg on an echocardiogram [5]. Likewise, Wong et al. found a significant association between subclinical right-ventricular dysfunction and increased BMI [6]. Both idiopathic and secondary forms of PH have been shown to be related to obesity in clinical research. A single-center study revealed that in obese subjects the prevalence of secondary forms of PH was 38% [7]. Similarly, data from the REVEAL registry analysis have suggested a higher prevalence of overweight and obese subjects among patients affected by idiopathic forms of PAH. Interestingly, the authors reported that most of the patients with higher BMI were in worse functional classes at enrollment [8]. These findings suggest that obesity may influence both PH onset and severity. 

The pathogenic mechanisms of obesity-driven PH are complex and heterogeneous. Firstly, adipose tissue is an endocrine system implicated in the regulation of vascular and metabolic homeostasis as well as in inflammatory response through the production of bioactive substances known as adipocytokines [9]. Inflammation has an important role in the pathogenesis of PH as confirmed by the perivascular infiltration of inflammatory cells and neo-lymphogenesis in the peribronchial areas observed in both human and animal models of PH [10,11]. Quarck et al. found an increased level of C-reactive protein (CRP) in PAH patients and described a strong association between this biomarker and disease behavior [12]. In particular, the authors reported that CRP was a predictor of mortality and clinical outcomes of PAH patients. This finding was also confirmed using a more sensitive inflammation marker as the highly sensitive C-reactive protein in a more recent study [13]. In obesity, the adipocyte dysfunction is characterized by abnormal secretion of endocrine and paracrine mediators as result of excessive body fat deposition. Several biological agents including interleukin-1 (IL-1), IL-6, tumor necrosis factor-α (TNF-α), adiponectin, adipsin, leptin, and visfatin contribute to chronic inflammation in obese subjects. Leptin and adiponectin are the two most relevant adipokines in PH since their involvement has been clearly demonstrated in the pathogenesis and severity of the disease, although the biological role of both proteins is still to be clarified. Leptin is a pro-inflammatory adipokine normally involved in glucose and lipid metabolism and usually responsible for a vasodilator effect [14]. However, in overweight and obese subjects, increased levels of leptin can cause endothelial dysfunction, changing its effect on vascular tissue and participating in the pathogenesis of the disease [15]. Recent research revealed that lower leptin levels in PAH patients, corrected by BMI, were associated with an increased overall mortality and the leptin/BMI ratio represented a negative predictive value for mortality at two years [16]. Furthermore, in a cohort of patients with scleroderma, elevated plasma levels of adipsin were also related to the development of PH, especially in the limited cutaneous form of systemic sclerosis. Interestingly, whereas elevated brain natriuretic peptide(BNP) was not significantly associated with PH defined by both echocardiogram and RHC (*p* = 0.91 and *p* = 0.23, respectively), higher adipsin levels were significantly associated with PH assessed by echocardiogram or RHC (*p* < 0.0001 and *p* = 0.001, respectively) [17]. Adipokines can also regulate the inflammatory state of the vasculature in PH patients; indeed, caveolin-1 and peroxisome proliferator-activated receptor-gamma (PPARγ) are expressed in adipose and vascular tissues and have an important role in metabolic and vascular homeostasis through an intense interaction with adipokines. 

Several other factors contribute to the pathogenesis of PH in obese subjects. Hyperuricemia, which has a high prevalence among overweight individuals, has been reported to be an independent risk factor for both primary and secondary forms of PH. Levels of circulating uric acid were related to both PH severity and mortality [18]. Indeed, chronic hyperuricemia causes a local flow reduction within the pulmonary vessels counteracting nitric oxide (NO) production and increasing levels of endothelin. The impaired pulmonary vessel regulation contributes to endothelial dysfunction and subsequently increases the pulmonary pressures [19]. Furthermore, triglyceride and free-fatty-acid deposition in the myocardium of obese subjects could lead to the development of a cardiomyopathy characterized by eccentric ventricular hypertrophy (dilation without wall thickening) and diastolic heart failure usually seen in severely obese patients. The increase of left-ventricular filling pressures associated with left-ventricular failure can determine a secondary form of PH [20]. Finally, hypoxic vasoconstriction and subsequent pulmonary arteriolar remodeling due to repetitive nocturnal hypoxemia, as occurs in obstructive sleep apnea (OSA), could cause right-ventricular hypertrophy and PH. The prevalence of OSA in the obese population has been reported at about 40% [21]. Likewise, obesity hypoventilation syndrome (OHS) is associated with more severe PH although the pathogenic mechanisms are not fully elucidated [20,22]. 

## 3. Biological Role of Adiponectin

Adiponectin is a protein hormone of 244 amino acids, encoded by the *APM1* gene (adipose most abundant gene transcript 1), and located on the long arm of chromosome 3 (3q27). It is commonly found in concentrations between 5 and 30 μg/mL in the blood serum accounting for 0.01% of total serum proteins. Adiponectin is synthesized as a monomer of 28–30 kDa mainly within the adipocytes. However, it is also expressed in human and murine osteoblasts, liver parenchyma cells, myocytes, epithelial cells, and placental tissue [23]. In adipocytes, the processes of adiponectin biosynthesis and secretion are modulated by several molecular chaperones in the endoplasmic reticulum, including: ERp44 (endoplasmic reticulum resident protein 44), Ero1-La (ER oxidoreductase 1-La), and DsbA-L (disulfide-bond A oxidoreductase-like protein) [24,25,26]. After post-translational adaptations, the protein is merged into multimeric forms including trimers, hexamers, and high-molecular-weight (HMW) oligomers [27]. Homotrimer—low molecular weight (LMW)—is the oligomeric adiponectin base block. The hexameric form of adiponectin arises from the formation of a disulfide bond between two trimers mediated by the free Cys39. Likewise, the hexameric form represents the building block for the HMW adiponectin, which comprises 12–18 hexamers [28]. From the full-length protein proteolysis is generated the globular adiponectin, a globular C-terminal domain of adiponectin, which has analogous biological activity [29]. Adiponectin acts through the interaction with transmembrane G-protein-coupled receptors: AdipoR1 (primarily expressed in skeletal muscle) and AdipoR2 (the most abundant form in the liver) (Figure 1). Both receptors have also been found in pancreatic β-cells, macrophages, endothelial cells, and smooth muscle cells and within atherosclerotic plaques [30,31]. AdipoR1/R2 expression may be influenced by physical training, which acts promoting the receptors’ mRNA production within the human skeletal muscle [32]. 

AdipoR1/R2 show distinctive affinity for the different forms of adiponectin. The C-terminus fragment of AdipoR1 has high affinity for globular adiponectin; conversely, AdipoR2 presents intermediate affinity for both the globular and full-length adiponectin. Both these receptors activate AMP-activated protein kinase (AMPK), p38 mitogen-activated protein kinase (p38 MAPK), and peroxisome proliferator-activated receptor alpha (PPARα) promoting fatty-acid oxidation and glucose uptake in murine models [33] (Figure 1).

Adiponectin hexamers and the HMW multimers also act as ligands for T-cadherin (CDH13) [33,34]. This receptor is a glycosylphosphatidylinositol-anchored extracellular protein lacking the intracellular domain. T-cadherin is highly expressed in the systemic and pulmonary vasculature. However, the mechanisms underlying the activation of adiponectin intracellular signaling are still not fully elucidated. In murine models, T-cadherin may exert essential pro-angiogenic functions and its deficiency has been shown to counteract tumor neovascularization, limiting tumor growth [34]. On the other hand, T-cadherin absence counteracts adiponectin-induced cardioprotective effects in both short- and long-term cardiac hypertrophy as well as in a myocardial ischemia–reperfusion injury [35]. A strict regulatory axis between T-cadherin and adiponectin has been proposed as T-cadherin-deficient mice develop hypoadiponectinemia; likewise, low T-cadherin tissue expression was observed in adiponectin-deficient mice [23].

A growing body of research has documented that adiponectin has pleiotropic effects on inflammation and cell proliferation [36,37,38,39,40]. Whereas most adipose-tissue adipocytokines typically promote pro-inflammatory pathways, adiponectin has been proved to have anti-inflammatory activities [41,42,43]. Many inflammatory cells including macrophages, NK cells, lymphocytes, as well as pulmonary epithelial cells present receptors for adiponectin on their surface [44]. In vitro studies on macrophages have shown that adiponectin inhibits the activation of the NF-κβ [44,45] molecular pathway, reducing the phagocytic activity and the production of Interferon gamma (INF-γ) and TNF-α [46,47,48] and also stimulating the differentiation of macrophages toward the M2 phenotype [49,50]. In addition, adiponectin increases the production of cytokines with anti-inflammatory activity in lymphocytes—such as IL-10 [47]—and has the ability to bind some circulating chemokines by inhibiting their pro-inflammatory action [24,51]. Adiponectin also acts at the level of the endothelial cells by inhibiting the NF-κβ mediated activation, reducing the expression of adhesion molecules [52]. An important alteration found at the level of the vascular structure in pulmonary hypertension is the obliteration of the lumen of the small arteries secondary to the migration of mesenchymal smooth muscle cells (MSMCs). The MSMCs migration signaling is activated by several growth factors, such as platelet-derived growth factor (PDGF), epidermal growth factor (EGF), and vascular endothelial growth factor (VEGF) [10,53,54,55,56]. Although a specific role for adiponectin in the regulation of growth factors is not described, several effects on vascular remodeling have been demonstrated [41,57,58,59]. In vitro, adiponectin suppresses the proliferation and migration of smooth muscle cells (SMCs) [60]. Similarly, in vivo, mice with adiponectin-secretion deficiency develop a pathological accumulation of SMCs at the level of small-caliber vessel walls of the pulmonary circle [57]. Finally, adiponectin also plays a role in the modulation of vascular tone. Adiponectin has a direct vasodilator action [61,62]. Experimental studies show that adiponectin-deficient mice develop systemic arterial hypertension [63]. 

## 4. Adiponectin Prevents Pulmonary Hypertension

Adiponectin may represent a potential molecular link between obesity and PH, since a protective role for the pulmonary vasculature has been proposed (Figure 2). Adiponectin knockout (KO) mice are more prone to endothelial dysfunction, as a result of impaired vasoreactivity [64]. The protective influence of adiponectin in PH is supported by both in vitro and in vivo experimental data. Direct and indirect mechanisms have been proposed as models of the action of adiponectin in PH. 

One of the direct mechanisms is represented by the ability of adiponectin to regulate the endothelial nitric oxide synthase (eNOS) and NO levels. Accumulating evidences suggest that adiponectin acts as an endogenous modulator of the endothelial-derived NO production [65]. Interestingly, in the murine hyperlipidemic model, Li et al. reported that treatment with the globular domain of adiponectin enhanced eNOS phosphorylation at Ser1177, but not its expression, attenuating hyperlipidemia-induced inducible nitric oxide synthase (iNOS) expression (*p* < 0.05). Consequently, although adiponectin slightly reduced the total NO production, it significantly increased ACh-induced vasorelaxation. In addition to this unique property, adiponectin reduced the superoxide overproduction observed in hyperlipidemic vessels (*p* < 0.01) [24]. Besides NO regulation, adiponectin also directly functions by suppressing vascular inflammation and proliferation. In vitro data have demonstrated that adiponectin directly counteracts inflammatory processes in human lung cell lines, strongly supporting the anti-inflammatory role of this adipokine in lung disorders [66]. Lung endothelium in adiponectin-deficient mice exhibits an increase in perivascular inflammatory-cell infiltration [67]. Likewise, the overexpression of adiponectin has been shown to protect mice from developing PH in response to inflammation and hypoxia attenuating the pulmonary vascular remodeling [68]. In the pulmonary vasculature, adiponectin promotes the direct inhibition of the pro-inflammatory mTOR and NF-κβ pathways; this results in a reduction of the perivascular inflammatory-cell infiltration. Finally, adiponectin has a direct effect on vascular smooth cell proliferation: In vitro studies revealed that adiponectin suppresses smooth muscle cell proliferation and migration. This biological activity is mediated by the inhibition of the AMPK activation and mTOR activity, which stimulate cell growth and proliferation. In PH patients, the mTOR pathway promotes the induction of growth factors such as PDGF, EGF, and fibroblast growth factor (FGF) leading to vascular smooth muscle cell proliferation [69].

Another important linkage between adiponectin’s biological activities and PH is insulin resistance (IR). In animal models, the association between IR and hypoxemia induces the development of PH [70]. IR has also been reported to be more prevalent in patients with PAH, impairing both mortality and hemodynamic parameters [71]. In addition, elevated levels of insulin have been demonstrated to decrease AdipoR1/R2 expression, limiting adiponectin’s biological activities, a phenomenon known as adiponectin resistance [72]. Consequently, hyperadiponectinemia could be the result of the adipose tissue’s response for preventing adiponectin functional resistance. In a case–control study, Santos et al. demonstrated higher circulating adiponectin values in PAH patients than in controls (12.4 ± 6.9 vs. 8.1 ± 4.5 μg/mL; *p* < 0.05). Conversely, no statistically significant differences were found in plasma levels of leptin, visfatin, and resistin between the groups [73]. Similarly in another study, adiponectin, leptin, and endothelin-1 levels in idiopathic pulmonary arterial hypertension (IPAH) patients’ serum were found to be higher than those in healthy volunteers. A significant correlation has been reported between serum adiponectin concentration and pulmonary vascular resistance (PVR) (r = 0.449, *p* = 0.009) [74]. Although serum adiponectin levels resulted increased in PH patients, within the endothelial cells, adiponectin was found to be reduced in patients with congenital heart disease with pulmonary hypertension (CHDPH). Zhou et al. observed that adiponectin levels were higher in healthy subjects than in patients with coronary heart disease(CHD) with or without PH. Furthermore, they demonstrated that adiponectin and NO augmented after three months of sildenafil, whereas, endothelin-1 (ET-1) levels declined [75]. 

The indirect mechanism of adiponectin action in PH seems to be mediated by the insulin-sensitizing transcription factor PPARγ. PPARγ is an established vasoprotective transcription factor since its activation suppresses smooth muscle cell proliferation and migration [76]. Many key genes involved in PH development are targets of PPARγ and its pharmacological activation is translated into anti-proliferative, anti-inflammatory, pro-apoptotic, and direct vasodilatory effects in the vasculature [77]. Adiponectin, interleukin-6 (*IL-6*), monocyte chemotactic protein-1 (*MCP-1*), and *eNOS* are some of the target genes activated by PPARγ upon ligand activation [78]. Interestingly, increased adiponectin production appears likely to be a significant mediator of the systemic effects of PPARγ activation; more importantly, in vivo, adiponectin is required for PPARγ-mediated improvement of endothelial function in diabetic mice [76]. Therefore, adiponectin not only acts as a protective factor in PH but also represents one of the molecular mediators responsible for the use of PPARγ activators as therapeutic molecules in the treatment of PH.

In addition, the hemodynamic effects of the left heart chambers on the pulmonary circulation may be mediated by adiponectin. In patients with chronic heart failure (CHF), increases in BNP are related to the production of adiponectin [79]. Furthermore, Sam et al. reported that in mice with aldosterone-induced cardiac remodeling, adiponectin contributes to the preservation of chamber size in diastolic dysfunction and diastolic heart failure. The authors demonstrated that total left-ventricle wall thickness was increased in adiponectin KO-aldosterone vs. wild type-aldosterone mice (1.03 ± 0.06 mm versus 0.85 ± 0.04 mm, *p* < 0.05) [80]. Interesting data are reported in PH Groups 3 and 4 also. In patients with sleep-disordered breathing (SDB), Domagała-Kulawik et al. found that the median serum adiponectin levels were significantly reduced in OSA patients compared with healthy subjects (*p* < 0.05). Interestingly, the lowest serum concentration was observed in obese OSA patients. Similarly, the adiponectin/BMI ratio (A/BMI) was lower in obese OSA patients than in the control group (0.26 (0.19–0.44) vs. 0.62 (0.41–0.99), *p* = 0.0006). Furthermore, the authors reported decreased A/BMI levels in non-obese OSA patients though the difference with healthy subjects in this group was not statistically significant (*p* = 0.08) [81]. 

Finally, in a cohort of patients with CTEPH, Isobe et al. demonstrated that serum adiponectin levels inversely correlated with several clinical parameters including 6 min walk distance (6MWT), mixed venous oxygen saturation, body mass index, body weight, insulin levels, triglycerides (TG), TG: high-density lipoprotein (HDL)-cholesterol ratio, homeostasis model assessment of IR (HOMA-IR), and tricuspid annular-plane systolic excursion. Positive correlations have been found between adiponectin and BNP concentrations, pulmonary vascular resistance (PVR), and mPAP [82]. In conclusion, adiponectin holds pleiotropic effects by both inhibiting inflammatory and proliferative endothelial pathways and enhancing the PPARγ-mediated activities. Adiponectin’s contribution to PH pathogenesis appears considerable regardless of the underlying causes of PH.

## 5. Adiponectin Modulation in the Treatment of Pulmonary Hypertension

Based on the biological properties of adiponectin resulting in vascular effects, the use of this adipokine in the therapeutic scenario of PH is currently under investigation (Table 1). 

In the murine model of PH, the exogenous administration of adiponectin by inhalational or intravenous route has shown efficacy in attenuating allergen-induced airway inflammation and hyper-responsiveness [83]. Analogously, the administration of adenovirus-adiponectin into the tail veins of mice exposed to hypoxia improves hypoxia-induced pulmonary arterial remodeling [84]. Additional therapeutic approaches include: Thiazolidinediones (such as pioglitazone and troglitazone) and synthetic ligands of PPARγ that lead, indirectly, to an increment in production and secretion of adiponectin resulting in the suppression of PASMC proliferation [85]. In the same way, fenofibrate, a PPARγ agonist, ameliorates endothelial functions by reducing levels of inflammation and increasing adiponectin [86,87]. Among antihypertensive drugs, renin–angiotensin system (RAS) blockers have been demonstrated to improve plasma adiponectin levels; specifically, they were more effective than amlodipine, doxazosin, and metoprolol [88,89]. Moreover, Schupp et al. examined the effects of different angiotensin type-1 receptor (AT1R) antagonists on PPARγ function in mouse cells showing that these agents may induce PPARγ activation. Furthermore, they demonstrated that the PPARγ activation was also observed even in the absence of AT1R suggesting a direct activity mediated by the AT1R blockers [90]. Mineralocorticoid receptor (MR) blockade in the murine model, increases adiponectin expression in adipose and cardiac tissue, downregulates p38 MAPK protein expression, and promotes PPARα protein expression which, sequentially, upregulates adiponectin-receptor mRNA expression [91,92]. Finally, promising results have been shown by adipose-derived stem cells (ADSCs) on PAH. Li et al. [86] investigated the effects of ADSCs coupled with adiponectin on PAH showing that the ADSCs–adiponectin administration via the jugular vein in mice attenuates the hemodynamic changes and vascular remodeling of PAH (mPAP: 21.44 ± 0.89 mmHg versus 32.62 ± 1.77 mmHg in the PAH group, *p* < 0.05). Furthermore, the combination of adiponectin and ADSC therapy improved the pulmonary hemodynamics more than ADSC therapy alone (mPAP: 21.44 ± 0.89 mmHg versus 27.11 ± 2.12, *p* < 0.05). Similar results have been observed in preventing PASMC proliferation via the AMPK/BMP/Smad signaling pathway [93].

In conclusion, the paucity of drugs targeting the pulmonary vascular dysfunction coupled with the promising results in the murine models suggests that the modulation of adiponectin levels may represent an intriguing opportunity in the PH therapeutic scenario. Nevertheless, more comprehensive research on molecular mechanisms and long-term clinical outcomes are strictly required before any speculation can be provided. 

## Figures and Tables

**Figure 1 ijms-20-00912-f001:**
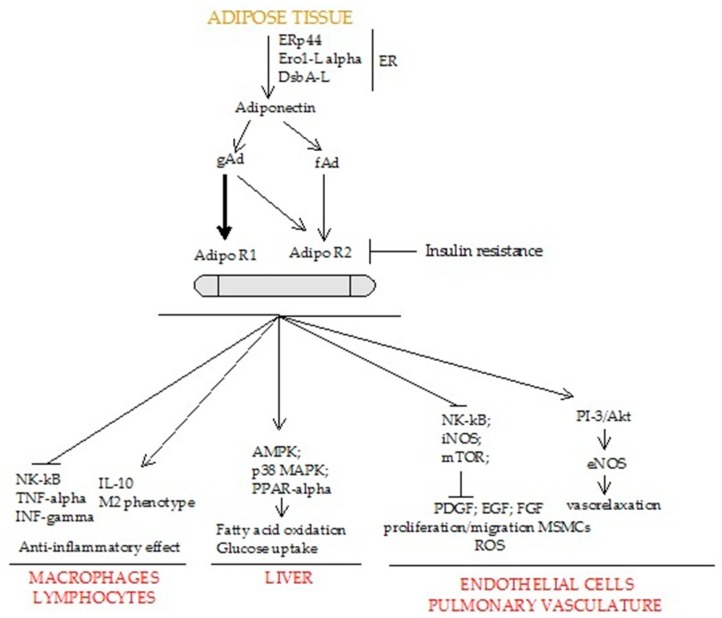
Cascade of cellular biological signaling activated by adiponectin–AdipoR1/R2 interaction. Adipo R1: Adiponectin receptor 1; Adipo R2: Adiponectin receptor 2; AMPK: Adenosine monophosphate–activated protein kinase; DsbA-L: disulfide-bond A oxidoreductase-like protein; EGF: Epidermal growth factor; eNOS: Endothelial nitric oxide synthase; ER: Endoplasmic reticulum; Ero1-L alpha: Endoplasmic reticulum oxidoreductin-1 (Ero1)-L alpha; ERp44: Endoplasmic Reticulum resident protein 44; fAd: full-length adiponectin; FGF: Fibroblast growth factor; gAd: globular adiponectin; iNOS: Inducible nitric oxide synthase; INF-gamma: Interferon gamma; IL-10: Interleukin 10; MSMCs: Mesenchymal smooth muscle cells; mTOR: Mammalian target of rapamycin; NF-kB: Nuclear factor -κB; PDGF: Platelet-derived growth factor; PI-3/Akt: Phosphoinositide-3-kinase/ Akt; PPAR-alpha: Peroxisome proliferator-activated receptor-alpha; PPAR-gamma: Peroxisome proliferator-activated receptor-gamma; p38 MAPK: p38 mitogen-activated protein kinase; ROS: Reactive oxygen species; TNF-alpha: Tumor necrosis factor-alpha.

**Figure 2 ijms-20-00912-f002:**
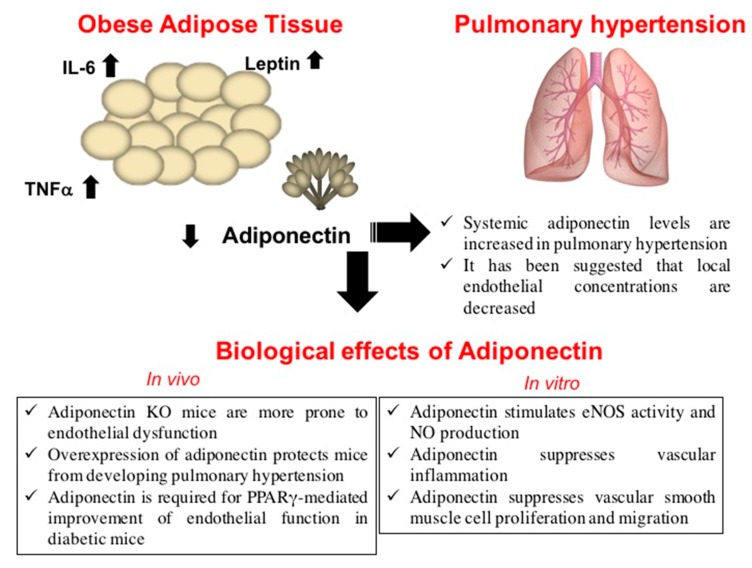
Effects of adipose-tissue hormones on endothelial cell homeostasis. eNOS: endothelial Nitric Oxide Synthase, IL-6: Interleukin-6, KO: knockout, NO: Nitric Oxide), PPARγ: Peroxisome Proliferator-Activated Receptor gamma, TNF-α: Tumor Necrosis Factor-α.

**Table 1 ijms-20-00912-t001:** Adiponectin modulation in pulmonary hypertension.

Molecule/Drug	Mechanism of Action	Authors
Thiazolidinediones	Synthetic ligands of PPARγ: ↑ production and secretion of adiponectin: ↓ PASMCs proliferation	Kubota et al. [85]
Fenofibrate	PPARγ agonist: ↑ levels of adiponectin: ↓ levels of inflammation ↑ endothelial functions	Li et al. [86], Xu et al. [87]
RAS blockers ARBs	↑ levels of adiponectin ↑ PPARγ activity	Yenicesu et al. [89] Schupp et al. [90]
MR blockade	↑ heart and adipose tissue adiponectin; ↑ PPARα protein expression; ↓ p38 MAPK; ↓ inflammation; ↓ insulin resistance	Wang et al. [92]
ADSCs and adiponectin	↓AMPK/BMP/Smad signaling pathway ↓ PASMCs proliferation	Luo et al. [93]

ARBs: Angiotensin type-1 receptor (AT1R) blockers; ADSCs: Adipose-derived stem cells; BMP: bone morphogenetic protein MR: Mineralocorticoid receptor; p38 MAPK: p38 mitogen-activated protein kinase; PPAR: Peroxisome proliferator-activated receptor; RAS: Renin–angiotensin system; PASMCs: Pulmonary arterial smooth muscle cells; ↑: Improve; ↓: Reduce.

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
