# Peer review of "Pulmonary Hypertension and Obesity: Focus on Adiponectin"

_ijms, 2019, doi:10.3390/ijms20040912_

Reviewer 1 Report

The relevance of this topic to pulmonologists and endocrinologists would be substantial. For the same reasons, the subject is also particularly suited to readers of this journal.

The title reflects the purpose of the manuscript well. In contrast, the abstract could provide a better summary of the body of the contents; it is too sketchy.

The overview is an excellent precis of the background.

The rest of the manuscript is extremely difficult reading.  There is much material, which is a strength of the manuscript. Simultaneously, the ideas and paragraphs do not flow well, and are disjointed.  It almost seems as if the literature on the subject was collected, and findings in each article were summarized in 1-2 sentences, then combined as best as possible. There is little direction in the paragraphs that reflects either enthusiasm or weighs the importance of what is expressed. Rather than a cohesive entity, the effect is staccato-like with abrupt starts and stops of ideas and facts found in each reference. In short, the translational development needs improvement with better transitions, but also stating why the various facts led to a certain mini-conclusion associated with that section.

In summary, although the individual biochemical/physiologic actions of adiponectin are nicely covered, their relative significance in the overall (medical) pathogenesis needs to be added.

A central schematic or graphic of the conceptual foundation with dependent and independent variables would be very helpful after the overview.  Table 1, appearing later on, is well done.

While the many effects of adiponectin could account for the associations observed, there should be some unified discussion of any particular predilection of these effects with the pulmonary vasculature rather than all vasculature. Are we to assume that since adiponectin is capable of these effects generally that it is a matter of chance that in some individuals the pulmonary tree will show sufficient changes to produce clinical disease?  Given these many fundamental actions, why do some individuals develop PAH and others, for instance, CHD, PAD or nephropathy?

Even though the extensive list of references is an asset, especially in this type of review, the average age is not recent, and might be improved.

Specifics, but similar minor items appear elsewhere:

Lines 17, 52 and other places:  eliminate the

Line 53 bode should read body

Line 76 and other places: plasmatic should be plasma

Line 152: eliminate “from”

Author Response

Accordingly, we have revised the abstract.

A possible explanation is linked to the intrinsic heterogeneity of the pulmonary hypertension disease. The different disease forms together with the complexity of adiponectin models counteract a linear narration. Anyway, in this version we have added transitions and sentences for easing the information flow. We included conclusions for each section in order to summarize.

In the revised version of the manuscript we improved both “Biological Role of Adiponectin” and “Adiponectin prevents from pulmonary hypertension” paragraphs remarking the clinical significance of adiponectin.

We thank the reviewer for the suggestion and accordingly we have added one more figure in the revised version of the manuscript in the attempt to summarize the information.

The role of adiponectin in many diseases with an inflammatory component is a complex matter of debate. In some inflammatory diseases, an increase of adiponectin levels has been described while in others the levels drop down. This difference may be attributed to the different environment present in the different conditions, to the involvement of the adipose tissue in the disease pathogenesis, to the types of inflammatory cells involved and other mechanisms. In addition, the cause-effect relationship between adiponectin and the disease is also often unclear. The most accredited hypothesis is that firstly there is the establishment of the PH disease that consequently lead to an increase of adiponectin levels in the attempt to exert anti-inflammatory actions.

Accordingly, we replaced old references with more recent ones.

Specifics, but similar minor items appear elsewhere:

Lines 17, 52 and other places:  eliminate the

Done

Line 53 bode should read body

Done

Line 76 and other places: plasmatic should be plasma

Done

Line 152: eliminate “from”

Done

Reviewer 2 Report

In this review paper, Perrotta et al. hypothesize an important role for adiponectin in pulmonary hypertension, and present several pieces of evidence in support of their thesis. This is a well written review. However, I think that some points need to be addressed to improve reliability and clarity of this paper.

1.       The Authors imply that in a recent study (Ref.16) “plasmatic levels of leptin were found significantly higher” in patients affected by scleroderma-associated pulmonary hypertension (page 2, line 77). However, that study shows that leptin levels were increased in patients with limited cutaneous scleroderma. That study links serum adipsin levels to scleroderma-associated pulmonary hypertension.

2.       The Authors talk about a link between PH and increased serum levels (page 4, line 157), but they do not show evidence of this link. They quote a study (Rhodes CJ et al, Ref.60), which evaluated plasma proteome in patients with PH, but did not find any association between adiponectin and PH. To the best of my knowledge, the only study that has found increased serum levels of adiponectin in patients with PH is the one by Kochelkova EA et al (Calcif Tissue Int 2017). This study found increased adiponectin in the serum of patients with idiopathic PAH. In contrast, Zhou C et al (Pharmazie 2016) found decreased levels of adiponectin in endothelial cells from patients affected by congenital heart disease with pulmonary hypertension.

3.       The Authors describe the affinity of different adiponectin receptors for globular and full-length adiponectin (page 3, line 116). However, they did not explain previously  to the readers that adiponectin can be present either as full-length protein or as the C-terminal residue alone (globular form).

4.       The Authors say that they show adiponectin receptors in a Figure (page 3, line 110). However, I cannot find any figures in the paper.

5.       In the Table, the Authors quote several studies on molecules used in PH by the first author’s name. For better clarity, the Authors should show the reference number of this studies.

6.       There are several formal errors in the paper (eg. “bode” instead of “body” at line 53, repeated “as T-cadherin” at line 128). Please revise the whole paper for formal errors.

Author Response

We apologize for the mistake. We didn’t intend to include data about adipsin in the manuscript. We clarified in the text and removed the wrong reference.

We thank the reviewer for the observation. We have added these studies and one more case control study showing reporting hyperadiponectinemia in PH patients.

We have extended the discussion on the different forms of adiponectin and in particular we have added information about the globular form.

The figure was included in the submission according to Editorial indications as a separate file. However, in this version we included the figure in the text.

We have changed the table adding the reference number.

We apologize for the mistakes. We have revised the grammar and corrected the formal errors in the whole paper.

Round 2

Reviewer 1 Report

The Authors have extensively modified their review paper, correcting some imprecisions and reorganizing their material in a more readable way.

I have only a few minor suggestions:

In Fig.2, there is an object between the adipose tissue and the lungs that I could not recognize. Please explain it to the readers.

If I have interpreted the Authors' intentions, Figure2 should stress adiponectin as a link between adipose tissue and pulmonary vasculature. However, adiponectin does not appear prominent in the Figure.

On this issue, I would add titles to the Figures, to highlight their role in the paper.

Throughout the paper, English could be improved.

Author Response

Dear Editor and Reviewers,

Many thanks for your help in improving the overall quality of the paper.

In the revised version, we made some changes to fig 2 and we have added a figure legend.

Some changes to English mistyping.

On Behalf of all Authors

Your Sincerely
Fabio Perrotta

Reviewer 2 Report

The revised submission has addressed this reviewers concerns.

The authors are to be congratulated on their contribution.

Author Response

Thank you!